# Characterization of Polyallylamine/Polystyrene Sulfonate Polyelectrolyte Microcapsules Formed on Solid Cores: Morphology

**DOI:** 10.3390/polym16111521

**Published:** 2024-05-28

**Authors:** Aleksandr L. Kim, Egor V. Musin, Yuri S. Chebykin, Sergey A. Tikhonenko

**Affiliations:** Institute of Theoretical and Experimental Biophysics, Russian Academy of Science, Institutskaya St., 3, 142290 Puschino, Moscow Region, Russia; kimerzent@gmail.com (A.L.K.); eglork@gmail.com (E.V.M.); kobepoftruth@gmail.com (Y.S.C.)

**Keywords:** polyelectrolyte microcapsules, morphology, size of microcapsules, thickness of shell, pores of microcapsules, pH, ionic strength, temperature, PSS, PAH, solid core

## Abstract

Polyelectrolyte microcapsules (PMC) based on polyallylamine and polystyrene sulfonate are utilized in various fields of human activity, including medicine, textiles, and the food industry, among others. However, characteristics such as microcapsule size, shell thickness, and pore size are not sufficiently studied and systematized, even though they determine the possibility of using microcapsules in applied tasks. The aim of this review is to identify general patterns and gaps in the study of the morphology of polyelectrolyte microcapsules obtained by the alternate adsorption of polystyrene sulfonate and polyallylamine on different solid cores. First and foremost, it was found that the morphological change in polyelectrolyte microcapsules formed on different cores exhibits a significant difference in response to varying stimuli. Factors such as ionic strength, the acidity of the medium, and temperature have different effects on the size of the microcapsules, the thickness of their shells, and the number and size of their pores. At present, the morphology of the microcapsules formed on the melamine formaldehyde core has been most studied, while the morphology of microcapsules formed on other types of cores is scarcely studied. In addition, modern methods of nanoscale system analysis will allow for an objective assessment of PMC characteristics and provide a fresh perspective on the subject of research.

## 1. Introduction

A polyelectrolyte microcapsule (PMC) is a modular, spherical-shaped structure that was first introduced in 1998 as a new type of nanoengineered multifunctional material [1,2]. They are prepared by the layer-by-layer deposition of oppositely charged polyelectrolytes on the surface of spherical templates (cores) ranging in size from 60 nm to several micrometers, followed by the dissolution of the core [3,4]. A schematic representation of this process is presented in Figure 1.

The step-by-step formation of the microcapsule shell allows its properties such as thickness, permeability, elasticity, stiffness, functionality, and others to be configured to meet specific requirements [5,6,7,8,9]. One of the most important features of capsules is their multifunctionality due to the possibility of modifying the shell with organic molecules [10], polymers [11], inorganic nanoparticles [12], carbon nanotubes [13], antibodies [14], etc. In addition, these capsules can be filled with a wide range of different molecules such as proteins/enzymes, drugs, nanoparticles, polyelectrolytes, fluorescent tags, and even bacterial spores [15,16,17,18,19,20,21].

The relative simplicity of their preparation and the possibility of controlling various properties will, in the long term, allow their use in many fields of human activity such as catalysis, agriculture, biotechnology, medicine, etc. [22,23]. In fact, they have great potential as containers for targeted drug delivery [24,25], the intracellular transport of small peptides [26], corrosion protection [27], biosensors [28], bioreactors [29], etc. As can be seen from the above, there are numerous studies devoted to the application of PMCs in various fields. However, the study of their physicochemical and mechanical properties, as well as the morphology of polyelectrolyte microcapsules, is necessary for the successful encapsulation of substances and the prediction of the influence of environmental conditions on the structure of microcapsules and on the encapsulated substances.

In particular, the size of the microcapsules, the thickness of their shell, and the size of the pores are the characteristics that determine the possibility of using microcapsules in the given tasks. The size/diameter of the microcapsules should not exceed the minimum diameter of the channel of the system in which the microcapsules are to be used.

For example, the minimum diameter of mammalian blood vessels can vary significantly. Therefore, the size of the PMCs should not exceed the diameter of the vessels to avoid occlusion [30]. The thickness of the shell affects the deformation and rupture of the capsule shell under an external load, which is a critical parameter for ensuring the protection and/or release of encapsulated materials in delivery systems, diagnostic systems, and smart materials [31]. The size of the pores determines the ability of substances of different sizes to pass through the shell of a polyelectrolyte microcapsule, both from the external environment into the microcapsule and vice versa [32]. Furthermore, it has been demonstrated that various physicochemical factors, including ionic strength, pH, temperature, and solvent polarity, can influence the morphological parameters of polyelectrolyte microcapsules [3,33,34]. The underlying molecular mechanisms of these effects remain poorly understood, despite their critical importance, given the necessity of considering a range of incubation conditions for different applications.

At present, the morphological characteristics of polyelectrolyte microcapsules obtained by the alternate adsorption of polystyrene sulfonate and polyallylamine on solid templates are the most extensively studied, but the results of these studies are fragmented and lack coherence. Therefore, the objective of this review is to collate and organize existing studies that investigate the morphology of polyelectrolyte microcapsules obtained by the alternating adsorption of polystyrene sulfonate and polyallylamine on solid templates.

## 2. Size and Shape of Polyelectrolyte Microcapsules 

The morphology of polyelectrolyte microcapsules, including the size and shape of the microcapsules, as well as the thickness, porosity, and density of their shell, is a key characteristic that determines the degree of capsule loading, release rate, and stability under different environmental conditions. These factors are crucial to consider when developing targeted delivery systems, diagnostic systems, sorption systems, and other applications.

It is essential to consider the size and shape of the PMC when designing polyelectrolyte microcapsules for certain applications. First of all, it is necessary to take into account the minimum channel diameter of the system in which the microcapsules will be utilized because the size/diameter of the microcapsules should not exceed this minimum diameter. For example, the minimum diameter of a human blood vessel is 8 μm, while that of a rat is 4 μm [30]. Therefore, depending on the object of study, it is necessary to create certain conditions for the formation of microcapsules of the required size. In addition, the size of PMCs significantly determines the adhesion behavior and the interaction of the capsule with environmental objects. In particular, their uptake by cells [35], efficiency of drug delivery [36], rate of removal from the bloodstream, and interaction with immune cells depend on the size of the microcapsules [37]. The shape of polyelectrolyte microcapsules is also a critical parameter as it affects the cellular processing of polyelectrolyte microcapsules [38] on the adhesion of capsules [39] and stability [40].

Therefore, depending on the object of study, specific conditions must be created for the formation of microcapsules of the required size. Moreover, the size of PMCs significantly influences the adhesion behavior and the interaction of the capsule with environmental objects. Specifically, factors such as their uptake by cells [35], the efficiency of drug delivery [31], the rate of removal from the bloodstream, and their interaction with immune cells are dependent on the size of the microcapsules [36]. The shape of polyelectrolyte microcapsules is also a crucial parameter as it affects the cellular processing of the capsules [37], their adhesion [38], and stability [39].

One of the key factors determining the size and shape of polyelectrolyte microcapsules is the type of core/template used for microcapsule formation. Meanwhile, various external factors can also influence their size, which also needs to be taken into account for their application across various fields.

### 2.1. Influence of the Core Type on the Size and Shape of the PMC

The size and shape of polyelectrolyte microcapsules obtained by the alternate adsorption of polystyrene sulfonate and polyallylamine are determined by the core used, as it sets the initial arrangement of the polyelectrolyte layers of PMC. Broadly speaking, cores can be classified into two types: hard and soft [8]. Soft cores are formed mainly on emulsion drops. However, capsules derived from such cores are characterized by high polydispersity in their size and low stability upon surface contact [41]. In this regard, solid cores are more commonly used for the creation of polyelectrolyte microcapsules. 

The most studied and widespread PMCs are microcapsules formed on melamine formaldehyde (MF) cores [1]. Polystyrene (PS) is also frequently used [2]. However, the complete removal of these cores requires solvents that are aggressive to biomolecules (pH below 1, THF, DMF, etc.). As a result, MnCO_3_ [6] and CaCO_3_ [42] cores which can be completely removed from PMC under relatively mild conditions (pH ≈ 6, EDTA, ethylene glycol tetraacetic acid, etc.) have been widely used [34]. Based on a literature analysis, the aforementioned cores are most commonly used in PMC preparation, making microcapsules based on these cores the most studied. In particular, the influence of MF, PS, MnCO_3_, and CaCO_3_ cores on the shape and size of polyelectrolyte microcapsules has been described in detail. Their images, obtained by scanning electron microscopy, are presented in Figure 2 [3,34,43,44].

As observed in Figure 1, MF, PS, MnCO_3_, and CaCO_3_ particles exhibit a regular spherical shape, or close to it in the case of CaCO_3_. Figure 2 presents a view of these particles, which significantly differ in surface roughness. The size of the polyelectrolyte microcapsules is dependent on the size and shape of the particle templates, which fall within the following ranges: melamine formaldehyde cores, 0.06 to 10 μm [4,45]; polystyrene cores, 0.1 to 16 μm [45,46]; MnCO_3_, 1.85 μm to 300 μm [47]; and CaCO_3_, 0.4 μm to 10 μm [47]. In contrast to PS and MF, CaCO_3_ and MnCO_3_ particles do not possess a perfect spherical shape with a smooth surface, but rather exhibit a rougher and more complex topography, which may influence the eventual shape and properties of the polyelectrolyte microcapsule. Subsequently, we will provide a detailed discussion on PMCs formed on different types of cores.

The size of polyelectrolyte microcapsules, formed on MF particles, can range from 60 nm to 10 µm [4]. According to Ljuljevic et al. [22], the size of the PMCs matches the size of the MF particles themselves (5.6 ± 0.2 µm), which is also confirmed in this study [1]. However, in the work of C.Gao et al. [23], the process of MF core dissolution was observed in real time using confocal laser scanning microscopy. It was shown that the capsules swell as a result of the MF core’s dissolution. The initial size of the PMC was 6.4 µm, which subsequently increased to 9.1 µm. Over time, the size of the capsule returned to its original state. The authors found that MF oligomers are too large to easily diffuse out of the capsules. The temporarily high concentration of dissolved MF oligomers inside the capsules creates osmotic pressure. This situation leads to the swelling of the capsules and exerts great mechanical stress on the polyelectrolyte film of the PMC shell. Moreover, this effect significantly increases with the increase in the core diameter [45].

PMCs formed on polystyrene cores exhibit similar characteristics. In the works of Christophe De’jugnat*, Gleb B. Sukhorukov, Heuvingh, and J. et al., it was observed that the size of PMCs did not differ from the size of the core after its dissolution [5,46,48]. Similar to the case with MF cores, the size of the PMC briefly increased before returning to its previous size. The reason for this increase is the same as with MF cores—the dissolution products of the polystyrene core are long polymer chains (Mw about 106 Da), which results in increased osmotic pressure inside the PMC.

In the case of MnCO_3_ serving as the core, the size and shape of the PMC coincide with the size and shape of the template [6]. However, similar to the previously described studies, no research has been conducted on the change in the microcapsule size during the dissolution of the core. Furthermore, comparative studies of polyelectrolyte microcapsules formed on melamine formaldehyde (MF) and manganese carbonate (MnCO_3_) particles, and on polystyrene (PS) and MnCO_3_ particles, were carried out in the works of Sukhorukov et al. [6] and Déjugnat, C. and Sukhorukov, G.B. [48]. According to the results of these studies, no significant difference was found in the morphology of the shells based on MF, PS, and MnCO_3_ particles. The size and shape of the prepared polyelectrolyte microcapsules correspond to the size and shape of the original templates.

However, the PMCs formed on a CaCO_3_ core exhibit significant differences. In the work of Volodkin et al. [42], CaCO_3_ particles of an approximately spherical shape, ranging from 4 to 6 μm in size and covered with a polyelectrolyte shell, were presented. After the core’s removal, the size and shape of the PMCs were preserved without a short-term increase in size. The authors suggested that the polyelectrolyte complex formed on the surface/inside of the capsule creates a strong yet porous network, protecting the capsule from high osmotic pressure and allowing Ca^2+^ ions to rapidly penetrate through the pores without causing destruction. The size of these pores can reach several tens of nanometers, aiding in maintaining an osmotic equilibrium. Additionally, the study [49] presented cross-sections of polyelectrolyte microcapsules formed on CaCO_3_ particles containing protein and on protein-free CaCO_3_ particles. The protein-free polyelectrolyte particles are complex internal structures consisting of multiple filamentous and enclosed nanoelements of a polyelectrolyte nature with a thickness of 20–30 nm. PMCs containing six to eight polyelectrolyte layers lack an outer shell. This structure is presumably due to the surface morphology of CaCO_3_ microparticles being very rough and comprising a large number of combined carbonate nanoparticles. The internal structure of CaCO_3_ microparticles is channel-like, with pore sizes in the range of tens of nanometers [42]. Polyelectrolyte microcapsules, which have more than nine polyelectrolyte layers, possess a formed shell [42], which confines the internal space of the microcapsule. As in the case of PMCs with fewer layers, this internal space is a polyelectrolyte complex with a complex channel-like structure.

In instances where the CaCO_3_ core contains protein molecules [49], polyelectrolytes are located only in the near-surface layer. The outer, spatially organized shell confines the inner volume, which is filled with a protein solution. This effect arises due to the “channels” of CaCO_3_ particles being filled with protein molecules [50], resulting in the adsorption of the polyelectrolyte on the outer surface of the spherulite.

Less commonly used templates for polyelectrolyte microcapsule formation include CdCO_3_ [34], SiO_2_ [51], and crystalline forms of substances that require encapsulation, such as fluorescein [52] and ibuprofen [19], among others.

The morphology of CdCO_3_ differs significantly from CaCO_3_ and MnCO_3_. The surface of CdCO_3_ is close to smooth while the shape of the core is close to cubic [34]. At the same time, no increase in the size of CdCO_3_ is observed when the core dissolves, while the cubic shape is preserved after its removal. 

The size of PMCs formed on SiO_2_ can vary from 0.03 to 100 μm [45], depending on the particle size. It can be presumed that the PMCs obtained on this template may increase in size after the removal of the core. In particular, Yang et al. demonstrated an almost twofold increase in the size of microcapsules after core removal [51]. However, due to the extremely small size of the SiO_2_ particles (200 nm) used in this study, limitations are imposed by the diffraction limit of resolution for optical microscopy, making this method unsuitable for the direct size estimation of PMCs. Therefore, transmission electron microscopy, which involves heating the sample during sample preparation, was used for size determination in the work of Yang et al. This resulted in the loss of about 65% of the polyelectrolyte mass.

From what has been written above, it can be concluded that the type of core used and its size are fundamental factors which the size and shape of polyelectrolyte microcapsules depend on. However, the size and shape of PMCs can be modified by external factors such as the pH of the medium, the ionic strength of the solution, and the incubation temperature, as well as by increasing the number of shell layers.

### 2.2. Influence of the Number of Layers on the Size and Shape of the PMC

The most obvious way to change the size of polyelectrolyte microcapsules is to increase the number of polyelectrolyte layers in the shell, as each subsequent layer of polyelectrolyte increases the thickness of the shell and, consequently, the size of the PMC. However, the addition of one layer increases the size of the PMC by a maximum of 20 nm. In several studies, it has been confirmed that the size of PMCs formed on PS and MF cores remains largely unchanged by the number of layers of the PMC shell [45,48]. A similar pattern is observed in the case of MnCO_3_ as a core [48,53].

### 2.3. Influence of the pH of the Medium on the Size and Shape of the PMC 

Another factor that can influence the size and shape of polyelectrolyte microcapsules is the pH of the solution.

In the work of Christophe De’jugnat* and Gleb B. Sukhorukov, it was shown that PMCs on a PS template increase in size in an alkaline medium, while the capsule size decreases when PMCs are incubated at low pH values [48]. The increase in size is presumably due to the deprotonation of the ammonium groups of PAH under alkaline conditions, which results in an increase in the size of PMCs due to the mutual repulsion of PSS. The decrease in the microcapsule size is presumably due to a greater number of PAH amino groups becoming positively charged, which enhances the interaction with the anionic sulfonate groups of PSS. As a result, the polymer chains shrink, leading to smaller capsules. 

However, the change in the size of PMCs at different pH values depends on the number of shell layers. In an acidic environment (pH ≈ 1), PMCs with fewer layers become significantly smaller than microcapsules with more layers. This is presumably because capsules with more layers have a stiffer shell, which prevents them from shrinking [46]. However, in an alkaline medium, the authors observed a different result with a similar nature of the phenomenon. Microcapsules containing 10 and 12 layers decrease in size in an alkaline medium by almost twofold (from pH 11.5), while if there are 14 and 16 layers, they conversely increase in size by twofold [48]. Capsules with fewer layers lack sufficient stiffness to resist shrinkage, while PMCs with more layers have a rigid shell [46]. However, this does not explain the effect where the PMC shell becomes smaller than the original core. Nevertheless, the authors conducted an experiment in which PMCs in an alkaline medium (pH > 11) with varying numbers of layers were incubated in 0.5 M NaCl. The authors hypothesized that the addition of salt would increase the shell’s flexibility, causing the PMCs to decrease in size. Indeed, according to the work of Heuvingh et al. [46], the capsule shell exhibits a pronounced softening with the increasing ionic strength of the solution, but only at salt concentrations from 3 M NaCl. Thus, it can be inferred that the increase in the softness and size of PMCs at 0.5 M NaCl may be due to the synergy of the effects of ionic strength and an alkaline medium. As a result, it can be inferred that the PMC shell becomes smaller than the original core due to the enhanced effect of ionic force on the PMC shell under alkaline conditions.

In the same paper, Christophe De’jugnat* and Gleb B. Sukhorukov [48] showed that PMCs formed on the MnCO_3_ core (3.6 µm) increase in size by a factor of three in an alkaline medium (pH 12.5). Moreover, this effect was observed regardless of the number of polyelectrolyte layers. The authors attributed this effect to the deprotonation of ammonium groups of PAH at an increasing pH. As a result, the polyelectrolyte shell is dominated by a negative charge due to PSS, which increases the electrostatic repulsion of polyelectrolytes and increases the mutual distance, leading to an increase in the size of the PMC and eventually to the complete destruction of the shell.

PMCs formed on MF or CaCO_3_ particles do not change their size at different pH values. This effect was demonstrated in the work of Gleb B. Sukhorukov et al. [54] where confocal microscopy images show that the size of PMCs at pH 10 and 3 does not differ from the size of the MF particles on which the microcapsules are formed. In the work of Changyou Gao et al. [23], the dissolution of the MF core at pH 1.1 was studied, and it was demonstrated that the particle size does not change during the entire time of core dissolution and the removal of its degradation products. A similar effect was demonstrated using CaCO_3_ as a template. In the work of Tong et al. [55], it was shown that the size of PMCs remains the same at both pH 2.5 and 7, which was also confirmed in Ref. [56]. However, at a pH above 12.5, capsule dissolution was observed within 3 s [46]. Meanwhile, at low pH values (below 1), the dissolution of capsules was also observed over time, over a period of 30 min [57].

Summarizing the above, it can be asserted that the size of polyelectrolyte microcapsules changes depending on the type of core and the pH of the solution: specifically, PMCs formed on MF and CaCO_3_ cores do not change their size at different pH values of the medium, and PMCs formed on PS cores decrease in size in an acidic medium and increase in size in an alkaline medium. In the case of PMCs formed on MnCO_3_, the size of the capsules increases in an alkaline medium; however, there is currently no data on the impact of low pH values on the size of such PMCs, which requires additional research in this area.

### 2.4. Effect of Ionic Strength on the Size and Shape of the PMC

In the work of Heuvingh et al., it was demonstrated that with an increasing NaCl concentration in the solution, a decrease in the size of PMCs formed on the PS core was observed [46]. A significant difference in size was noted at salt concentrations of 3 M and higher, with a maximum reduction of 20%. It was also shown that the incubation time of PMCs in a medium with a high ionic strength does not affect the size change. After incubating the PMCs in a salt solution for 30 min, the size stabilizes, and even if the PMCs are incubated in a 5 M NaCl solution for 3 weeks, the size remains the same as when incubated for 30 min. Additionally, this process is irreversible. Re-incubating PMCs in pure water maintains the size that was present when incubated in salt. The authors suggested that the addition of salt leads to a softening of the ionic interactions between the polyelectrolytes forming the multilayer structure, thereby allowing the molecular rearrangement and annealing of the capsule structure. Furthermore, the authors have shown that the salt anion, rather than the cation, has a greater impact on the size change of the PMC. While the SO_4_^2−^ ion has a much smaller effect (11% reduction) than Cl^−^, NO^3−^ has a stronger effect on the morphology change (38% reduction). This effect of reducing the size of PMCs is consistent with the position of anions in the Hofmeister series, in which ions are classified according to their lyotropic properties, i.e., their ability to salt in or salt out proteins. Given that some amino acids are characterized by electrostatic interactions in proteins similar to the inter/intra polyelectrolyte interaction, it can be assumed that the structural influence of anions of the Hofmeister series on the polyelectrolyte complex of the PMC shell will have a similar character. This also supports the assumption of a decrease in the size of the PMC with an increasing salt concentration due to changes in the structure of the multilayer shell.

Similar observations on the effect of different salts on the morphology of polyelectrolyte capsules were independently made by Georgieva et al. for the PAH/PSS system formed on the MF core [58]. However, the concentration of salts used was 1.7 M, which does not allow us to draw a definite conclusion about the influence of NaCl and ionic strength on the size of PMCs since at this concentration, the decrease in the size of PMCs is only 2%. At the same time, in the work of Lebedeva et al. [59], it was shown that the size of PMCs formed on the MF core does not change at NaCl concentrations from 0 to 5 M. Thus, we can conclude that the effect of salt on the size of PMCs formed on the MF and PS core is different.

When considering PMCs formed on the MF core containing PSS, there was also a decrease in the size of the PMCs observed with the increasing ionic strength, but this decrease differed from that of microcapsules formed on the PS core. In particular, Lebedeva et al. showed [59] that PMCs formed on the MF core, filled with PSS, significantly increase in size compared to “empty” PMCs. This is due to the high osmotic pressure caused by the encapsulated PSS [60]. Increasing the concentration of NaCl in the solution leads to a decrease in the diameter of these PMCs up to a concentration of 3 M, after which the size of the microcapsules does not change. However, even in a 5 M NaCl solution, the size of the “filled” PMCs does not reach the size of the “empty” capsules. Presumably, this effect is due to the excessive osmotic pressure caused by the counterions of the encapsulated PSS.

The behavior of PMCs formed on CaCO_3_ cores in solutions with different ionic strengths is significantly different compared to the PMCs described above. In the work of Pechenkin et al., it was shown that PMCs morphologically do not change in the presence of NaCl salt in solution, even at a concentration of 6.1 M NaCl [57]. Thus, it can be assumed that the decrease in the size of PMCs obtained on an MF or PS core is due not only to the rearrangement of the shell, but also to the influence of external osmotic pressure on the PMC shell, which leads to its reduction. Meanwhile, in PMCs formed on a CaCO_3_ core, this shell is absent and as a result, there is no change in size in the presence of salt. Another possible explanation could be differences in the shape of the shell; the matrix structure of the capsules formed on CaCO_3_ may have sufficient stiffness to resist shrinkage.

### 2.5. Influence of the Temperature of the Medium on the Size and Shape of the PMC 

Kim et al. suggested that the PMCs of a PSS/PAH composition formed on MF cores decreased slightly (less than 10%) when incubated at 80 °C for 5 h, which is comparable to the measurement error [61]. However, Leporatti et al. showed that polyelectrolyte capsules of the composition (PSS/PAH) decrease in size upon heating [62]. In addition, the authors studied the effect of temperature on the size of PMCs at different MF particle sizes, with different outer layers of PSS or PAH, and in the presence or absence of heating the acid solution used to remove the core. As a result, it was shown that the parameters described above did not significantly affect the reduction of the PMCs, finding that, on average, the PMCs were reduced by approximately 15% at 70 °C. This reduction caused by heating was attributed to the transition to a more coiled arrangement of polymers in the multilayers due to the breaking and reforming of ion pairs, resulting in a more compact arrangement of polyelectrolyte molecules. Prior to heating, the size and shape of the capsule were maintained by charge bonding, and the breaking of this bond upon heating caused shrinkage. A similar effect was observed by Kohler et al. at a higher temperature of 120 °C [3], whereby the PMCs decreased by more than 50 percent.

At present, there are no studies investigating the size of PMCs formed on the PS cores at different incubation temperatures. However, a number of studies have applied thermal effects on this type of microcapsule in order to achieve applications [63,64]. Based on the results of these studies, it can be inferred that changing the temperature does not significantly affect the size of the microcapsules. However, these studies do not directly address the effect of temperature, which limits the ability to draw specific conclusions regarding this aspect. Consequently, further studies specifically aimed at investigating the effect of temperature on the microcapsule size are needed for a more complete understanding.

In the case of PMCs formed on CaCO_3_, the work of A. V. Dubrovskii et al. found that the capsules decrease in size with an increasing temperature and heating time, regardless of whether they consist of an even or odd number of layers [65]. When microcapsules were incubated for 20 min at 90 °C, a decrease in size was observed. The PMCs with an even number of layers ((PSS/PAH)_6/8/10_) decreased by 28% and those with an odd number of ((PSS/PAH)_7/9_) layers decreased by 24%, respectively. In the case of polyelectrolyte microcapsules formed on CaCO_3_ + protein spherulites, a decrease in capsule size was also observed. However, in this case, the influence of the parity/oddity of the polymer microcapsule layers had an opposite effect. Microcapsules with an even number of layers ((PSS/PAH)_6/8/10_) reduced by 26% and those with an odd number of layers ((PSS/PAH)_7/9_) reduced by 34%, respectively. It should be noted that in these types of PMCs, unlike microcapsules formed on CaCO_3_ spherulites without protein, polyelectrolytes are located only in the capsule shell, limiting its internal volume filled with protein solution, although some of the protein may be bound to the shell itself. This difference is probably due to the presence of the shell in these capsules, in contrast to the matrix arrangement of polyelectrolytes throughout the internal volume of capsules formed on a CaCO_3_ core.

According to the above description, it can be concluded that polyelectrolyte microcapsules formed on different types of cores significantly differ in size changes in response to different physicochemical stimuli. When MF and CaCO_3_ were used as cores, polyelectrolyte microcapsules did not change their size at different pH values of the medium, whereas PMCs formed on PS cores decreased in an acidic medium and increased in an alkaline medium with subsequent disintegration (which is also characteristic of PMCs on MnCO_3_). Given that, morphologically, PMCs formed on MF and PS cores do not differ, we can hypothesize that the reason for the difference in the size change of PMCs and the pH of the medium lies in the process associated with the removal of the cores. In particular, the work of Christophe De’jugnat* and Gleb B. Sukhorukov [48] has already demonstrated the influence of tetrahydrofuran, a solvent used to remove the templates of PMCs formed on PS cores, on the stability of PMCs formed on MnCO_3_. After 15 h of incubation in this solvent, the shell stability of PMCs formed on MnCO_3_ was significantly increased in an alkaline medium. In addition, oligomers formed as a result of the incomplete dissolution of the MF cores may remain in the PMC shell, which may affect its mechanical properties and, as a consequence, the size change under the influence of the solution’s pH [23].

Also, the ionic strength of the solution does not affect the size of PMCs formed on MF and CaCO_3_ cores, but most likely the mechanisms of the lack of influence may be different. In the case of using PS cores, the PMCs decrease at ionic strengths above 3 M. The differences in PMCs obtained on MF and PS cores can be explained in the same way, as in the case of the influence of the solution’s pH. However, the nature of this phenomenon may be different and requires separate study.

In the case of temperature treatment, the polyelectrolyte microcapsules formed on MF cores and CaCO_3_ cores are characterized by a decrease in size, which is also characteristic of microcapsules obtained on PS cores; however, due to the lack of work aimed at studying this particular effect, we cannot state this unequivocally. Nevertheless, we can observe a general tendency to decrease the size of all presented types of capsules during temperature treatment, which may be directly related to the interpolyelectrolyte interaction between polyallylamine and polystyrene sulfonate, in which polyelectrolytes are arranged more compactly relative to each other.

## 3. Thickness of the Shell of Polyelectrolyte Microcapsules

The thickness of the PMC shell is one of the most important parameters of microcapsule morphology because the thickness of the shell determines the deformation and rupture of the capsule shell under the action of external load, which is extremely important to ensure the protection and/or release of encapsulated materials. In particular, the thickness of the shell of the PMC primarily affects the stability of the microcapsule during core disruption. If the shell is thinner than a certain value, it will be destroyed by osmotic pressure. In addition, depending on the application and the physicochemical or mechanical conditions, capsule disruption may be desirable or preventable. Therefore, an understanding of the conditions that affect the thickness of the PMC shell is essential in the design of PMCs.

### 3.1. Influence of the Number of Layers on the Thickness of the PMC’s Shell

The number of layers of the PMC shell that adsorb onto the core directly affects its thickness. The literature data are presented in Table 1.

Table 1 shows that the shell thickness at the same number of layers does not differ significantly in the case of using MF, MnCO_3_, or PS as a core. Moreover, the shell thickness of the PMC increases linearly with the increasing number of polyelectrolyte layers, with a thickness increment of 3–3.8 nm per layer. It is noted in Dong et al. that the minimum thickness of the PMC shell (formed on the MF core) should not be less than 10 nm because it will lead to its destruction under the action of osmotic pressure in the process of removing the core from the microcapsule [60]. Considering that PMCs formed on MF, MnCO_3_, and PS cores have an extremely similar morphology and level of shell deformation under the action of osmotic pressure as a result of the removal of these cores [5,6,48,66], it can be assumed that the behavior of these capsules will be similar for shell thicknesses less than 10 nm.

Polyelectrolyte microcapsules formed on CaCO_3_ cores differ significantly. PMCs containing six to eight polyelectrolyte layers are devoid of an outer shell. Such a structure was presumably formed due to the complex internal channel-like organization of CaCO_3_ microparticles [42]. PMCs that have more than nine polyelectrolyte layers possess a shell that is about 46 nm thick [49]. However, Volodkin et al., in their study, measured the thickness of the polyelectrolyte complex that is formed on the CaCO_3_ core (without removing the core). The thickness of the PMC shell on CaCO_3_ is (six layers) 95 nm, with a single layer thickness of 8 nm, which is 3–5 times larger than in the case of PMCs on smooth surfaces [42]. The authors suggested that polyelectrolyte adsorption on the very rough surface of the CaCO_3_ microcore favors the formation of such a thick and porous polyelectrolyte network for the capsule wall. Obviously, the polyelectrolyte complex inside the capsule also has a significant share in the increase in thickness, but it is not possible to assess this. However, it is worth noting that when the cores are removed, there are many processes that affect the thickness of the shell, which does not allow us to compare PMCs with a removed core and with a non-removed core.

In the case of PMC formation on CaCO_3_ particles containing protein, the microcapsules form a pronounced shell. The authors have shown that the thickness of the shell increases with the increasing number of polyelectrolyte layers [49]. Six-layer microcapsules possess a fragmented outer shell with a thickness of 31–46 nm, while in seven-layer microcapsules, the thickness ranges from 43 to 49 nm. Starting from eight polyelectrolyte layers, the shell has a constant thickness of 60–70 nm.

### 3.2. Influence of pH, Ionic Strength, and Temperature of the Medium on the Thickness of the PMC’s Shell

As described above in Section 2.1, the thickness of the PMC shell, as well as its size, can be influenced by the pH environment, ionic strength, and temperature of the medium. However, there are currently no studies on the influence of the pH environment, ionic strength, and temperature of the medium on the thickness of the PMC shell formed exclusively on the MF core.

Georgieva et al. conducted a study on the impact of the solution’s ionic strength on the shell thickness of PMCs that were formed on the MF core. The findings demonstrated a 30–40% increase in shell thickness, as a function of salt (LiCl, NaCl, KCl, and CsCl), given a constant ionic strength (I = 1.67) [58]. In contrast, research by Lebedeva et al. indicated that the shell thickness of PMC increases by 10–15% with an increase in the NaCl concentration up to 2.5 M [59]. However, any further increase in the salt concentration results in a 40% reduction in the thickness of the PMC shell. It is postulated that these effects occur because, at a NaCl concentration up to 2.5 M, PSS and PAH are shielded from each other by the electrolyte’s counterions. Moreover, a substantial increase in the salt concentration leads to a significant rise in the dissociation of polyelectrolytes from the shell.

The ionic strength is one of the main factors of the shell thickness not only after core removal, but also in the process of shell formation around the template. In particular, Dong et al. showed that the ionic strength at which the PMC shell is formed, prior to core dissolution, also affects the thickness of the wall of the formed capsule [60]. In particular, at the lower salt concentration of 0.25 M, the shell thickness is 1.5–2 times thinner compared to the standard concentration of 0.5 M. However, it is worth noting that the minimum thickness at which the PMC shell is stable is the same in both conditions and is 10 nm. 

Studies devoted directly to the influence of the pH of the medium on the shell thickness of PMCs formed on MF cores are absent in the literature. However, Georgieva et al. studied the effect of 1.67 M Na_2_HPO_4_ (I = 10 M) at pH 8.72 and 1.67 M NaH_2_PO_4_ (I = 10 M) at pH 3.92 on the shell thickness of such PMCs. As can be seen, the concentrations of the salts correspond to the same ionic strength, but the pH of the solution differs. As a result, an increase of 30% was shown at pH 8.72, which corresponds to an increase in shell thickness from the ionic strength of the solution. However, at the same ionic strength but at pH 3.92, the shell thickness of PMCs increases by only 13% [58]. Accordingly, it can be assumed that the smaller increase in the shell thickness can be attributed to the low pH of the solution, which is associated with an increase in the [H^+^] concentration, which leads to an increase in the number of charged amino groups of PAH and, consequently, increases the interaction with PSS.

The influence of temperature on the thickness of the PMC shell, as presented in the literature, is exclusively based on the example of microcapsules formed on MF cores. Estrela-Lopis et al., for instance, demonstrated that maintaining a temperature of 70 °C for 4 h resulted in an approximately 38% increase in the thickness of the capsule wall [67]. This finding was corroborated by Leporetti et al., who reported a 32% increase in shell thickness [62]. An even more significant effect was demonstrated by Kohler et al., whereby PMCs incubated at 120 °C showed an initial shell thickness of 14.4 ± 1.4 nm, which increased more than tenfold to 160 ± 12 nm after heating [3]. The authors proposed that the thickening of the capsule wall was due to a rearrangement of the macromolecular layer constituents. This process would require the disruption and recombination of ionic pairs formed between charged groups of polyanions and polycations, which are crucial for layer stability. Consequently, the polymers that form the layer would adopt a conformation that is more entropically favorable.

## 4. Porosity of the Shell of Polyelectrolyte Microcapsules

Nanoscale pores/holes primarily determine the permeability of the shell of polyelectrolyte microcapsules. Currently, there are no definitive data confirming the existence of pores or other openings in the PMC shell. Typically, electron microscopy methods, which require PMC preparation through heating to temperatures above 300 °C or the use of organic solvents, are utilized to measure their sizes. This preparation process may lead to the formation of breakage. However, it is widely accepted that a decrease in the polyelectrolyte interactions of the PMC shell due to exposure to varying environmental conditions could result in the formation of pores/holes. These pores/holes could enable the penetration of high-molecular-weight substances. 

The shell structure of polyelectrolyte microcapsules formed on an MF core varies with the number of layers and shell thickness. For a thickness of 12 to 16 nm, a mesoporous PMC shell structure with a large pore size was observed by Dong et al. Conversely, for shell thicknesses ranging from 16 to 28 nm, the shell structure is microporous [60]. ‘Mesoporous’ capsules have an average pore size of 2 to 50 nm, presumably formed by the breakthrough release of MF oligomers with an average size of 2–3 nm. The ‘microporous’ PMCs have an average pore size of less than 2 nm, which is likely the result of the trapping of MF oligomers by the multilayer shell obtained during core removal. According to previous studies, up to 20% (by weight) of the released MF remains in the shell [68].

For PMCs formed on a CaCO_3_ core, the lack of a distinct shell complicates the description of pore presence in this structure. Porous CaCO_3_ microparticles, measuring 4.5–5.0 μm, have a surface area of 8.8 m^2^/g and an average pore size of 35 nm [42]. Approximately 59% of the particle volume consists of solid CaCO_3_, while around 41% comprises the internal pore volume. This structure allows small molecules of PSS and PAH, as small as a few nanometres in size, to penetrate the interior through the pores during successive polyelectrolyte treatment, thereby forming a polyelectrolyte complex. It is suggested that the channel-like structure of the CaCO_3_ microcore, with its substantial internal volume, promotes the formation of a dense and porous polyelectrolyte network in the capsule wall, featuring numerous holes larger than 100 nm.

Macroscopic properties such as permeability, elasticity, etc., are strongly dependent on the environment: the temperature, solvent, ionic strength, and pH of the solution. Among others, the pore size of the shell of polyelectrolyte microcapsules depends on these parameters.

### 4.1. Influence of pH of the Medium on the Porosity of the PMC’s Shell

In particular, Antipov et al. studied the influence of the acidity of the medium on the permeability of the shell of PMCs formed on the MF core [15]. However, the authors also demonstrated the influence of the acidity of the medium on the pore size of the microcapsule shell. The obtained images are presented in Figure 3 [15].

Figure 3 shows that when the microcapsules are incubated in the pH 3.5 solution, holes as small as 100 nm are formed. In addition, the authors proved the reversibility of this process. When the same microcapsules are subsequently incubated in pH 10.5 (as well as at pH 12), these holes are absent.

However, the authors suggest that the presence of holes in the SFM preparation (dried during preparation) does not necessarily imply their presence in the capsules in the solution. It is likely that at a low pH, damage to the shell walls occurs, potentially leading to the formation of holes during drying due to layer shrinkage. This clearly indicates a weakening of the polyelectrolyte bond, which could increase permeability [15]. Moreover, numerous studies have explored the encapsulation of substances into this type of capsules by altering the solution’s acidity. For instance, Sukhorukov et al. incorporated FITC-dextran 75 kDa into PMC at pH 3. Above pH 8, FITC-dextran could not penetrate the capsule shell [54]. Similarly, Tiurina et al. encapsulated the proteolytic enzyme, achymotrypsin, in hollow PSS/PAH shells. The encapsulation was conducted at pH 4.0, where, according to the authors, pores formed in the capsule walls. Subsequent washing at pH 8.0 sealed the shell walls [69].

Halozan et al. demonstrated that (PAH/PSS)_8_ microcapsules, prepared on PS templates, were sensitive to a pH shift to the alkaline region [70]. Above pH 11.2, the polyelectrolyte network in the microcapsule wall undergoes changes in its structure and porosity. This pH region may correspond to the formation of pores in the wall due to the segregation of the polymer network, which is associated with the significant deprotonation of PAH. It is well established that the degree of protonation of a weak polyelectrolyte is highly dependent on its environment, particularly on the nature of the polyelectrolyte with which it is bound. Above pH 11, the degree of protonation of the PAH-PSS complex is virtually zero [33]. The fully ionized and negatively charged PSS causes the capsule wall to swell due to repulsion between its charged monomeric groups. This mechanism differs from the encapsulation mechanism at an acidic pH in (PSS/PAH) microcapsules templated on melamine formaldehyde particles [15]. As the pH decreases, the charge density on PAH increases, leading to partial polymer degradation and increased shell permeability [71]. Oligomers resulting from the incomplete dissolution of the MF core can also protonate at a low pH, contributing to the increase in permeability [23].

The most unusual and unexpected is the response to acidity of the PMC medium formed on the CaCO_3_ core. Pechenkin et al. demonstrated that the remaining CaCO_3_ in the capsules, after core removal, dissolves in a strong acid (0.1 M HCl or more concentrated), forming CO_2_, which is expected to diffuse out of the capsule. However, lowering the pH of the solution leads to the annealing of the polyelectrolytes (reducing their mutual interaction), resulting in the rearrangement of the polyelectrolyte complex by carbon dioxide bubbles. Consequently, this process decreases the CO_2_ diffusion rate and leads to the formation of CO_2_ bubbles inside the capsule. The bubble grows, causing the capsule to swell and its structure to change from matrix-like (completely filled with polymers) to shell-like (with a well-defined shell and hollow inner lumen). As the capsule expands, the shell thins, and when it reaches a critical size, shell permeability increases due to the formation of defects and the escape of CO_2_. The internal pressure within the capsule then decreases, causing the capsule to shrink back to its original size. Once the defects in the shell are restored, the capsule begins to inflate again. These oscillating inflate–deflate cycles repeat 4–5 times with a decreasing amplitude until most of the CaCO_3_ has dissolved or the capsule shell’s defects can no longer be repaired. Eventually, the capsule collapses [57].

From the above, it can be inferred that the acidity of the medium affects the pore size of polyelectrolyte microcapsules formed on different templates in various ways. In an acidic medium, the pores of the PMC shell on an MF core increase in size, but close in an alkaline medium. Conversely, for PMC on a PS core, the pores increase in size or number in an alkaline medium. Given that the microcapsules formed on the MF matrix have the same polyelectrolyte composition as those derived from polystyrene, it is possible that the increase in porosity under alkaline pH conditions is due to factors other than the chemical properties of the polyelectrolyte shell. This could include the spatial organization of the polyelectrolytes, the effect of an incompletely removed core, or its solvent. Meanwhile, microcapsules formed on a CaCO_3_ core are not directly affected by the pH environment.

### 4.2. Influence of Ionic Strength and Temperature on the Porosity of the PMC’s Shell

At present, there are no studies devoted to exploring the impact of ionic strength and solution temperature on the pore size of PMCs. However, if we hypothesize that the primary mechanism influencing the permeability of the microcapsule shell is the alteration in pore size, we can extrapolate from the results that focus on the effect of ionic strength and solution temperature on the permeability of PMCs.

For instance, Ibarz et al. investigated the effect of the solution’s ionic strength on the penetration of fluorescently labeled PAH through the shell of PMCs formed on the MF core. They demonstrated that at 10^−3^ M NaCl, M PAH ± rho(rhodamine) (70 kDa) could not penetrate the shell [72]. However, increasing the salt concentration above 10^−2^ M NaCl enabled the labeled polymer to pass through the shell. The authors proposed that this effect is due to an increase in pore size resulting from the higher ionic strength of the solution. On the other hand, Georgieva et al. suggested that PMC permeability can be attributed both to the presence of pores or defects in the capsule wall, formed during the preparation process, and to electrostatic interactions between the penetrating molecules and the charged surface of the capsule [73].

In the case of PMCs formed on a PS particle, Halozan et al. [70] demonstrated that the addition of 0.2 M NaCl more than doubled the amount of substance (PAH) incorporated into the PMC. However, the authors suggest that in this instance, the ionic strength of the solution did not result in an increase in the pore size of the shell. Instead, it led to a decrease in the hydrodynamic radius of the polyelectrolyte (PAH) from 8 nm in pure water to 5 nm with the addition of 0.1 M NaCl, which facilitated the penetration of more polyelectrolyte.

In the study conducted by Ibarz et al., a significant decrease (three-fold) in substance release from the PMCs formed on the MF core was observed after heating them at 80 °C for 30 min. This is presumably associated with a decrease in the pore size in the PMC shell [74]. However, it should be noted that the effect of the temperature alters the permeability of PMCs on a consistent basis.

### 4.3. Influence of Solvent Type on the Porosity of the PMC’s Shell

The influence of solvent remains a relatively underexplored area. In their work, Kim et al. studied the effect of solvent on the shell of PMCs formed on an MF core. They demonstrated that both the total shell pore area and pore size are dependent on the solvent used. For example, the total pore area was estimated to be approximately 1% for capsules immersed in water, and 1% and 4% for capsules immersed in ethanol/water and acetone/water mixtures, respectively. The analysis of pore size distribution in water indicates that the majority of the pores have a radius of 5–10 nm. No significant changes in pore size were found for capsules treated with ethanol/water mixtures. However, capsules treated with acetone/water mixtures exhibited a distinct change in pore size distribution, with the relative number of pores with a radius less than 10 nm decreasing by a factor of two. This was accompanied by the emergence of large pores with extremely irregular, non-circular shapes, with the average radius of large pores exceeding 40 nm [75]. This effect was utilized in the work of Lvov et al. to encapsulate urease in polyelectrolyte microcapsules [20].

## 5. Density of the Shell of Polyelectrolyte Microcapsules

Currently, the densities of the PMC shell are determined by mathematical calculations. In their studies, authors determine the shell’s thickness and volume using methods such as small-angle neutron scattering [67,76], transmission electron microscopy (TEM), and scanning force microscopy (SFM) [3], among others. However, to date, there is no method that allows for the direct determination of shell density, necessitating the use of several assumptions. This has been acknowledged by the authors in their respective studies.

One such assumption is the belief that the properties of the PMC shell are independent of the type of core used. As mentioned earlier, several studies use data (such as shell thickness) derived solely from PMCs formed on the MF core [5,61,76,77,78,79], even though the authors themselves may use polystyrene or other cores.

Another assumption is that the calculation of the PMC shell’s density does not account for the influence of the sample preparation procedure on the shell’s thickness. However, the use of high temperatures (up to 60 °C), organic solvents, and other environmental conditions can significantly impact the PMC shell, its thickness, and consequently, the calculation of its density.

In these studies, the effect of various environmental conditions such as the temperature, ionic strength, and pH of the solution on shell density does not consider the change in the mass of the PMC shell due to these factors. However, there are several studies [80,81,82] that have demonstrated the loss of polyelectrolytes by the PMC shell depending on different environmental conditions. Therefore, given that mass is one of the key parameters of density, such an assumption may not be entirely accurate.

Also, under various environmental conditions, as mentioned earlier [15,20,54,57,69,70,72,73,74], the number and size of pores in the shell of polyelectrolyte microcapsules change. However, these parameters are not taken into account by the authors in their studies. Nonetheless, the formation of a hole in the shell of a PMC can lead to an increase in shell density or in its thickness (which also affects the density calculation), provided that the mass of the shell remains constant.

In summary, given the issues described above, it can be concluded that there is a current need for new methods to determine the density of the PMC shell that are free from these limitations. Additionally, for a more accurate determination of the PMC shell density under various environmental conditions using standard methods, it is essential to consider the impact of these conditions on shell mass loss and pore formation.

## 6. Conclusions

Based on this review, it can be concluded that the morphology of PMCs is most extensively studied on microcapsules that are formed on MF cores. There are considerably fewer studies on the morphology of microcapsules formed on other types of cores. However, most applied research often uses the results of studies obtained for polyelectrolyte microcapsules formed on a melamine formaldehyde core to explain the influence of environmental conditions or to calculate the physicochemical characteristics of PMCs formed on different cores. This approach is not entirely correct, as this review has shown that significant differences were found in the changes in microcapsule size, shell thickness, and pore size in response to physicochemical stimuli.

When considering the influence of the medium’s pH on the size of PMCs, it is observed that PMCs formed on MF and CaCO_3_ cores do not change their size at different pH values of the medium. However, PMCs formed on PS cores decrease in size in an acidic medium and increase in size in an alkaline medium, followed by subsequent destruction (a characteristic also observed in PMCs on MnCO_3_). In the case of exposure to ionic solution strength, a similar lack of influence on the size of PMCs formed on MF and CaCO_3_ cores was observed, though the mechanisms of the phenomenon may differ. PMCs formed on PS cores are characterized by a decrease in size at ionic strengths above 3 M. Temperature effects lead to a reduction in the size of all types of capsules presented in this work, regardless of the template used.

In the context of studying the influence of the core type on the shell thickness of polyelectrolyte microcapsules (PMCs), it can be noted that no significant differences are observed for the same number of layers (this applies to MF, MnCO_3_, and PS cores). However, the thickness of the PMC shell increases linearly with the increasing number of polyelectrolyte layers. The polyelectrolyte microcapsules formed on a CaCO_3_ core, whose shell is formed after nine polyelectrolyte layers, are significantly different, with a thickness of about 46 nm. In the case of PMCs formed on CaCO_3_ particles containing protein, the microcapsules form a pronounced shell already after the sixth layer. However, the calculated thickness of one layer is twice that of the thickness of one layer of the shells of PMCs formed on MF, PS, or MnCO_3_ cores.

It has also been shown that the acidity of the medium affects the pore size of polyelectrolyte microcapsules formed on different templates in varying ways. In an acidic medium, the pores of the PMC shell on the MF core become larger, and in an alkaline medium, they close. In contrast, for PMCs on the PS core, the pores increase in size or number in an alkaline medium. Microcapsules formed on a CaCO_3_ core, however, are not directly affected by the pH of the medium. Despite these observations, it is not currently possible to make any assumptions about the mechanisms of this phenomenon, as there are no studies on the influence of other stimuli (ionic strength and temperature). This highlights the need for further research in this area, which may elucidate the impact of pore formation in the PMC shell on its permeability.

Separately, we would like to note that some studies conducted earlier require the use of more advanced methods for analyzing nanoscale systems, which will allow a more accurate and objective assessment of the characteristics of PMCs without the harsh conditions of sample preparation: sample heating, the use of organic solvents, and other conditions. Modern methods, such as Tunable Resistive Pulse Sensing (TRPS) technology [83] nano-impact electrochemistry [84], single-particle inductively coupled plasma mass spectrometry [85], focused ion beam nanoslice tomography [86], and others, allow us to avoid these drawbacks and adequately estimate the size of polyelectrolyte microcapsules (PMCs), the thickness of the shell, and the size of their pores. These approaches will not only allow researchers to clarify the already obtained data, but will provide an opportunity to evaluate new aspects of PMC behavior under different conditions to develop more effective methods of PMC application in various fields, ranging from medicine to electronics.

## Figures and Tables

**Figure 1 polymers-16-01521-f001:**
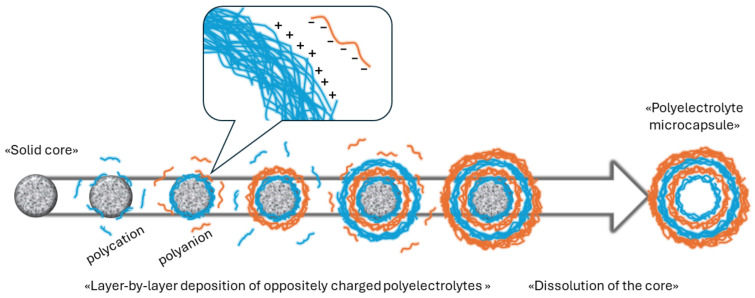
Scheme of the preparation of polyelectrolyte microcapsules.

**Figure 2 polymers-16-01521-f002:**
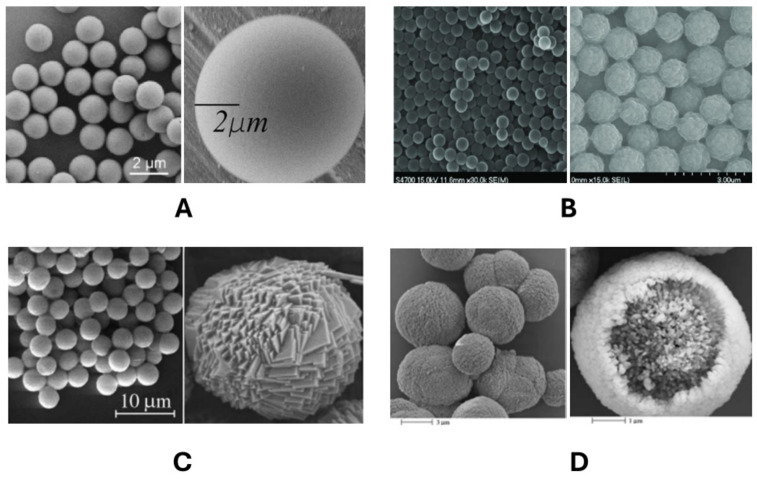
SEM images of (**A**) melamine formaldehyde [3,43]; (**B**) polystyrene [44]; (**C**) MnCO_3_ [34]; and (**D**) CaCO_3_ [34] particles. Adapted with permission from Refs. [3,34,43,44].

**Figure 3 polymers-16-01521-f003:**
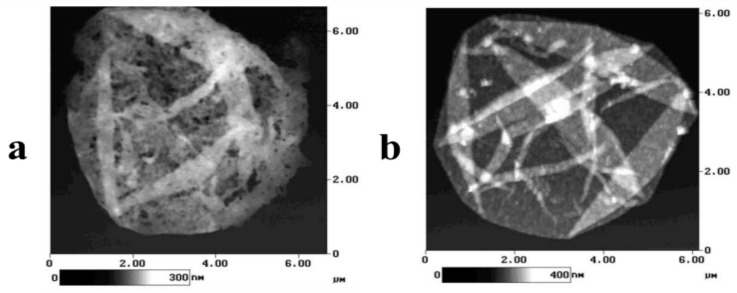
SFM images of polyelectrolyte microcapsules: (**a**)—pH 3.5; (**b**)—pH 10.5 [15]. Adapted with permission from Ref. [15].

**Table 1 polymers-16-01521-t001:** PMC’s shell thickness depending on the type of core.

Number of Layers	Types of Core									
	MF [60]		PS [48]		CaCO_3_ [42,50]		CaCO_3_ + Protein [42,50]		MnCO_3_ [48]	
	Shell Thickness, nm	Estimated Thickness of One Layer, nm	Shell Thickness, nm	Estimated Thickness of One Layer, nm	Shell Thickness, nm	Estimated Thickness of One Layer, nm	Shell Thickness, nm	Estimated Thickness of One Layer, nm	Shell Thickness, nm	Estimated Thickness of One Layer, nm
6	-	-	-	-	No shell		36.92± 3.28	6.15 ± 0.5	-	-
8	20 ± 2	2.5 ± 0.25	-	-	No shell		58.43 ± 2.82	7.3 ± 0.3	-	-
10	32 ± 3	3.2 ± 0.4	33 ± 2	3.3 ± 0.2	41	4.1	61.54 ± 3.21	6.15 ± 0.3	-	-
12	42 ± 1	3.5 ± 0.1	44.4 ± 2.4	3.7 ± 0.2	-	-	-	-	45.8 ± 4.7	3.8 ± 0.4
14	47 ± 2	3.35 ± 0.15	47.6 ± 2.7	3.4 ± 0.2	-	-	-	-	-	-
16	59 ± 3	3.69 ± 0.2	62.4 ± 3.1	3.9 ± 0.2	-	-	-	-	47.3 ± 2.9	3.5 ± 0.3
18	66 ± 1	3.67 ± 0.05	-	-	-	-	-	-	-	-

## Data Availability

Not applicable.

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
