# Peer review of "Characterization of Polyallylamine/Polystyrene Sulfonate Polyelectrolyte Microcapsules Formed on Solid Cores: Morphology"

_polymers, 2024, doi:10.3390/polym16111521_

Round 1

Reviewer 1 Report

Comments and Suggestions for Authors

Dear Authors,

The manuscript summarized and critically reviewed the existing literature that investigate the morphology of polyelectrolyte microcapsules obtained by adsorption of polyallylamine and polystyrene sulfonate on different solid cores. The figures and tables are with consistent of the main text. The conclusion is logical and derived from the content of the article. The authors also outlined future prospects to broaden investigation of morphology of microcapsules on cores different from melamine formaldehyde (most studied). The cited literature is up to the point.

The following remarks should be addressed to authors.

1. Did the authors receive a permission to reproduce the pictures in figures 2 and 3? I believe it is so and it should be mentioned in the captions.

2. There are some technical errors in lines 362, 383,384,511, 552, 606.

3. The text in lines 715-726 is repeated in lines 727-738 –needs correction

Author Response

We are grateful to the reviewer for the positive feedback on our work and hope it will be of interest to readers. We are happy to answer all your questions:

  1. Yes, before the publication of the article, we received permissions from the respective publishers to use the images. We appreciate you pointing that out to us. Changes have been made to the captions of the pictures.
  2. Thank you for pointing out these shortcomings in our work, they have been corrected.
  3. We grateful to your noticing the repetition of text that occurred during the translation of the article into English. Corrections have been made.

Reviewer 2 Report

Comments and Suggestions for Authors

The reviewed work discusses the morphological properties of polyelectrolyte microcapsules under various environmental conditions. The scope of this article is very extensive, but it focusses exclusively on microparticles composed of synthetic polymers. The manuscript is informative and well written. I have a few minor comments that the manuscript could benefit from if the authors follow my suggestions.

1. From the introduction and section 2, I conclude that the authors are referring to microparticles intended for intravenous administration. If there are other potential applications of these microparticles, they should be adequately addressed in the article.

2. In section 2.3. the authors switch the discussion from the core properties to the shell properties. I suggest organizing this section better so as not to confuse the reader.

3. The authors refer to various characteristics of microparticles and provide a similar explanation in separate sections, and thus it may seem somewhat chaotic in its current state. The overall perception of the manuscript's content would benefit from a concise conclusion to each section.

4. In lines 518 – 527 the authors refer to submicron particles, unlike the others discussed in the manuscript i.e. micropartciles, or even nanoparticles. It should be clearly stated that the morphological characteristics of submicron particles are significantly different from those of nanoparticles.

5. The information provided is repeated many times, i.e. lines 56 – 61 and lines 88 – 91, lines 137 – 140 and lines 145 – 147, etc.

Author Response

Dear Reviewer,

We thank you for your comments on our article, which will allow us to improve it and simplify its perception for the reader. We are pleased to respond to all your questions:

  1. Other examples of the use of polyelectrolyte microcapsules, taking into account their sizes, are in the text, and they were indicated in lines 93-99.
  2. Thank you for this comment, we have added a paragraph transition before section 2.3.
  3. Conclusions have been added in the section where it was necessary. In some sections, the authors refrained from adding conclusions at the end, as these sections touch on areas that are poorly studied and do not allow for a full conclusion, and the need for additional research is postulated at the beginning of the section.
  4. In lines 518-527, we do not consider nanoparticles and submicron particles, this chapter is about micron sized CaCO3 particles, which have pores, the size of which allows macromolecules such as PSS and PAA to penetrate, which as a result form an interpolyelectrolyte complex in the pores.
  5. In writing the text, lexical repetitions were deliberately used to focus the reader's attention on a specific element of the text. In addition, this technique simplifies the creation of semantic relationships between sentences in the text and reduces the likelihood of reader misunderstanding when reading the text (10.61425/wplp.2013.07.1.28; https://doi.org/10.1177/00472816231172904). We understand that not all scientists in the academic environment have a positive attitude towards the use of lexical repetitions, which has even been separately studied (https://doi.org/10.15476/ELTE.2015.180). However, the authors are convinced that the use of lexical repetitions will maintain a balance between academic quality and reader friendliness.